

# Aqueous chemical bleaching of 4-nitrophenol brown carbon by hydroxyl radicals; products, mechanism and light absorptivity

Bartłomiej Witkowski[1]*, Priyanka Jain[1] and Tomasz Gierczak[1]

[1]University of Warsaw, Faculty of Chemistry, al. Żwirki i Wigury 101, 02-089 Warsaw, Poland

*Corresponding author e-mail: bwitk@chem.uw.edu.pl

**Abstract**

The reaction of hydroxyl radicals (OH) with 4-nitrophenol (4-NP) in the aqueous solution was investigated at pH=2 and 9. As a result, the molar yield of the phenolic products was measured to be 0.20 ±0.05 at pH=2 and 0.40±0.1 at pH=9. The yield of 4-nitrocatechol (4-NC) was higher at pH=9; at the same time, a lower number of

phenolic products was observed due to the hydrolysis and other irreversible reactions at pH>7. Mineralization investigated with total organic carbon (TOC) technique showed that after 4-NP was completely consumed approx. 85% of the organic carbon remained in the aqueous solution. Hence, up to 65% of the organic carbon that remained in the aqueous solution accounted for the open-ring non-phenolic products.

The light absorptivity of the reaction solution between 250 and 600 nm decreased as a result of OH reaction with

4-NP. At the same time, 4-NP solution showed some resistance to chemical bleaching due to the formation of the light-absorbing by-products. This phenomenon effectively prolongs the time-scale of chemical bleaching or 4-NP via reaction with OH by a factor of 3-1.5 at pH 2 and 9, respectively. The experimental data acquired indicated that both photolysis and reaction with OH can be important removal processes of the atmospheric brown-carbon from the aqueous particles containing 4-NP.



# 1 Introduction

Atmospheric brown carbon (BrC) is a subfraction of organic aerosols (OA) that is characterized by strong, wavelength-dependent absorption of the electromagnetic irradiation in the near ultraviolet (UV) and visible (VIS) regions (Laskin et al., 2015; Yan et al., 2018). BrC is primarily produced by biomass burning (BB) and has a

negative impact on the local air quality and human health (Laskin et al., 2015; Yan et al., 2018). Due to the high UV-Vis absorptivity, BrC greatly contributes (up to 50%) to the radiative forcing of OA (Cordell et al., 2016; Zhang et al., 2017; Lu et al., 2015; Wang et al., 2014; Feng et al., 2013; Yan et al., 2018). Numerous organic compounds contribute to the atmospheric BrC (Laskin et al., 2015; Hettiyadura et al., 2021; Li et al., 2020a; Fleming et al., 2020); at the same time, a significant fraction of BrC chromophores remains poorly characterized

(Bluvshtein et al., 2017; Laskin et al., 2015).

Nitrophenols are widespread nitroaromatic compounds that been identified among the major chromophores of atmospheric BrC (Harrison et al., 2005b; Laskin et al., 2015; Bluvshtein et al., 2017). 4-Nitrophenol (4-NP) is one of the most atmospherically abundant and environmentally widespread nitrophenols (Harrison et al., 2005b; Laskin et al., 2015) and is characterized by very high absorption cross-sections in the UV-Vis region (Jacobson, 1999).

For these reasons, 4-NP has been identified as one of the major BrC chromophores (Xie et al., 2019; Mohr et al., 2013; Bluvshtein et al., 2017). 4-NP is present in the air (2006; Mohr et al., 2013; Belloli et al., 1999; Jacobson, 1999), rainwater (2006; Jaber et al., 2007; Nistor et al., 2001; Harrison et al., 2002), surface waters and snow (Balasubramanian et al., 2019; Vanni et al., 2001), soil (Webber and Wang, 1995), as well as in the atmospheric particulate matter (PM) (Liang et al., 2020; Kahnt et al., 2013; Kitanovski et al., 2020; Vione et al., 2009). Large

quantities of 4-NP are produced by the combustion of fuels, mainly biomass (Xie et al., 2019; Desyaterik et al., 2013; Mohr et al., 2013), coal (Li et al., 2020b; Liang et al., 2020), and also by diesel engines (Inomata et al., 2015). Moreover, 4-NP is introduced into the environment as an industrial waste (Aleboyeh et al., 2003; Behnajady et al., 2006; Behki and Khan, 1991). Several studies have confirmed, that 4-NP has an adverse impact on human health (Majewska et al., 2021; Rosenkranz and Klopman, 1990), it is a treat to aquatic organisms (Tenbrook et al.,

2003; Howe et al., 1994), and contributes to the decline of forests (Natangelo et al., 1999; Rippen et al., 1987).

Formation, chemical processing and decomposition (bleaching) of BrC can occur in air as well as in the atmospheric aqueous particles, which can involve direct photolysis and reactions with hydroxyl radicals (OH) (Hems et al., 2020; Laskin et al., 2015; Jiang et al., 2021; Li et al., 2020a; Forrister et al., 2015; Moise et al., 2015). Due to the high value of Henry's law constant (Sander, 2015), and high solubility in water, 4-NP can readily

partition into the atmospheric aqueous particles (Vione et al., 2009; Harrison et al., 2005b), Chemical and photochemical reactions in the atmospheric aqueous phase contribute to the formation (Harrison et al., 2005a;



Vione et al., 2003; Heal et al., 2007), functionalization (Vione et al., 2009; Vione et al., 2005) and decomposition (removal) (Braman et al., 2020; Harrison et al., 2005a) of 4-NP. The chemical and photochemical processing (aging) in the aqueous phase results in the change of the light absorptivity of aqueous particles containing 4-NP

(Zhao et al., 2015; Zhang et al., 2003; Braman et al., 2020). However, the interplay between light absorbance and chemical composition of the aqueous solution of 4-NP that has been subjected to photolysis and oxidation by OH is poorly characterized (Zhao et al., 2015; Zhang et al., 2003).

Aqueous reaction of 4-NP with OH (reaction 1) is known to produce aromatic compounds, including hydroquinone (HH), 1,2,4-trihydroxylbenzene (1,2,4-THB), 4-nitrocatechol (4-NC) and 4-nitropyrogallol (4-NPG) (Tauber et

al., 2000; Oturan et al., 2000; Zhang et al., 2003; Biswal et al., 2013; Daneshvar et al., 2007).

$$OH + 4\text{-NP} \rightarrow products \qquad\qquad (1)$$

However, the yields of the substituted phenols produced by reaction (1) remain ambiguous(Tauber et al., 2000; Zhang et al., 2003; Oturan et al., 2000; Daneshvar et al., 2007; Ding et al., 2016). Reaction (1) was previously

investigated on a molecular level but almost exclusively in the context of wastewater treatment via advanced oxidation processes (AOP) (Tauber et al., 2000; Zhang et al., 2003; Oturan et al., 2000; Ding et al., 2016) where the reaction conditions cannot be considered as atmospherically-relevant (Daneshvar et al., 2007; Zhang et al., 2003; Tauber et al., 2000; Ding et al., 2016). Consequently, it is currently difficult to evaluate whether or not reaction (1) is a relevant source of atmospheric BrC (Xiong et al., 2015; Oturan et al., 2000; Biswal et al., 2013;

Tauber et al., 2000; Kavitha and Palanivelu, 2005).

In the atmospheric aqueous particles, which are characterized by a broad pH-range (Herrmann et al., 2015), 4-NP (pKa≈7.2) can exist in both protonated and deprotonated forms (Rived et al., 1998). At the same time, little information is available about the pH impact on the products distribution of reaction (1) (Tauber et al., 2000; Oturan et al., 2000). There is no data available about the pH-dependence of the light absorbance of 4-NP solution

that has been subjected to the oxidation by OH (Zhao et al., 2015; Biswal et al., 2013). It should be also noted that the UV-Vis absorptivity of 4-NP and its oxidation products is strongly pH-dependent (Braman et al., 2020; Biswal et al., 2013).

The goal of this work was to investigate mechanism of OH reaction with 4-NP in the aqueous phase in context of atmospheric BrC formation and processing. Hence, the reaction (1) was investigated at 298 K in the aqueous phase

under acidic (pH=2) and basic (pH=9) conditions using the photoreactor developed in our laboratory:

Additionally, the phenolic products of reaction (1) were analyzed together with the changes in the UV-Vis absorptivity of the reaction solution (Witkowski et al., 2019). Phenols under investigation were quantified using



gas chromatography coupled to mass spectrometry (GC/MS). A possible mineralization and formation of volatile products was monitored with the total organic carbon (TOC) analyzer. The UV-Vis absorptivity of the reaction

solution as well as the molar absorptivity (base-e $\varepsilon$, $mol^{-1} \times L \times cm^{-3}$) of the phenols under investigation were measured between pH 2 and 9.

## 2 Experimental section

Materials and reagents used are listed in section S1 of the electronic Supplementary Information (SI).

### 2.1 Aqueous phase photoreactor

The aqueous phase photoreactor was described previously (Witkowski et al., 2019), and more details are provided in section S4.1. The reaction vessel was quartz jacketed reaction flask with 100 ml internal volume. All experiments were carried out at 298 K; the temperature of the reaction solution was maintained with a circulating water bath (SC100-A10, Thermo Fisher Scientific). Two lamps (TUV TL 4W, peak emission 254 nm, Philips) were used to irradiate the solution.

### 2.2 Experimental procedure

The reaction mixture was a 100 ml solution of 4-NP (concentration 100-250 μM) in deionized (DI) water. The pH of the solution was unbuffered or it was adjusted to pH 2 or 9 using HCl, $HClO_4$ and $Na_2HPO_4$ (50 mM) to investigate oxidation of fully protonated and deprotonated 4-NP (section S2). Hydrogen peroxide (concentration 5 mM) was photolyzed with UV irradiation (254nm) to generate OH, the estimated steady-state concentration of

OH was $1.4 \times 10^{-9}$ M (section S3) (Tan et al., 2009). Under these conditions 4-NP was almost completely consumed by OH within 1h. Aliquots of the reaction mixture were sampled every 5 min and analyzed by GC/MS, UV-Vis spectroscopy and TOC analyzer. The experimental procedure is described in detail in section S4.1.

### 2.3 Gas chromatography coupled with mass spectrometry

Analyses were carried out using GC-MS-QP2010Ultra gas chromatograph (Shimadzu) coupled with the

quadrupole QP-5000 mass spectrometer (Shimadzu), the instrument was equipped with AOC-5000 autosampler. Analytes were separated using capillary column ZB-5MSPlus (Phenomenex). The mass spectrometer was equipped with the electron ionization source (EI, 70 eV) and was operating in the selected ion monitoring (SIM) mode. For quantitative analyses GC/MS was calibrated with the standard solutions of 4-NP, HH, 1,2,4-THB, 4-NC that were identified as products of 4-NP reaction with OH. 2-Nitrophloroglucinol was used as a surrogate



standard for quantification of 4-nitropyrogallol (4-NPG) and 5-nitropyrogallol (5-NPG) identified among the

reaction products. Phloroglucinol was not identified among the product of reaction (1) (Xiong et al., 2015; Zhao

et al., 2013) hence, it was used as an internal standard (IS). Phenols were derivatized with acetic anhydrite (AA)

and analyzed via GC/MS (Regueiro et al., 2009). Detailed description of the analytical procedure is provided in

section S4.2.

**2.4   UV-Vis spectrophotometry**

UV-Vis measurements were carried out with i8 dual-beam spectrophotometer (Envisense) in 4 ml cuvettes with a

1 cm absorption pathlength. The absorbance of each aliquot of the reaction solution was measured between

wavelengths 230 and 600 nm. The pH of each sample taken from the reactor was adjusted between 2 and 9 (by

intervals of 1, see section S4.3) by adding a small amount of NaOH or $H_3PO_4$ solution in DI water. In a separate

set of experiments, the wavelength-dependent absorption cross sections, $\varepsilon$, (base-e; $mol^{-1} \times L \times cm^{-1}$) were

measured for 4-NP, HH, 1,2,4-BT, 4-NC and 2-NPG between are reported in appendix 1 (Fig. S5).

**2.5   Total organic carbon analysis**

Non-purgeable organic carbon (NPOC) was quantify with TOC-5050A analyzer (Shimadzu) connected to the ASI-

5000A autosampler (Shimadzu). The 1.5 ml of the reaction solution was diluted with the 1.5 ml of DI water. Then,

50 µl of 2M HCl was added via the TOC autosampler and each sample was sparged with oxygen for 2 min before

injection. The injection volume was 20 µl and each sample was injected into the instrument three times. The TOC

analyzer was calibrated with the standard solutions of 4-NP in DI water with concentrations between 3 and 35

$mg_{TOC} \times L^{-1}$; the squared linear coefficient of determination for calibration curve ($R^2$)=0.9995 was obtained.

**2.6   Light absorptivity and atmospheric lifetimes**

Production of light-absorbing compounds following reaction (I) was evaluated via eq. I.

$$\frac{\left(\int_{250nm}^{600nm} A_{R.mix}(pH)\right)_t d\lambda}{\left(\int_{250nm}^{600nm} A_{4-NP}(pH)\right)_t d\lambda} = \left(\frac{[4-NP]_0}{[4-NP]_t}\right)^{K_{abs}} \quad (I)$$

Where: $A_{R.mix}$ and $A_{4-NP}$ are integrated absorbance peak areas between 250 and 600 nm for the reaction mixture

measured between pH 2 and 9 at different time intervals (t), $[4-NP]_0$ and $[4-NP]_t$ are initial (0) and intermediate (t)

concentrations of 4-NP measured with GC/MS, and absorbance ($A_{4-NP}$) of 4-NP was calculated with Beer-Lambert



law using the ε measured in this work between pH 2 and 9. The expression described using eq. I follows the

function $y = A \times x^K$ (section S8).

The atmospheric lifetimes of BrC was evaluated by deriving the empirical $k_{bleaching}$ rate coefficients ($M^{-1}s^{-1}$) – eq. II.

$$k_{bleaching=} \ k_{OH}(4-NP) \times \frac{K_{A.rmix}}{K_{A.4-NP}} \ (II)$$

In eq. (II) the $k_{OH}$ is the bimolecular reaction rate coefficient ($M^{-1}s^{-1}$) for the reaction of 4-NP or 4-nitrophenolate with OH(Biswal et al., 2013; García Einschlag et al., 2003), $k_A$, $k_{A.rmix}$ are the first-order disappearance rate ($min^{-1}$) of the integrated absorbance peak for the 4-NP and for the reaction mixture, respectively (Fig. S9). The $k_{A.rmix}$ showed little dependence on pH at which the absorbance was measured, thus average values were used.

The organic carbon based mass-absorption coefficients ($MAC_{Int}$) of the reaction mixture were calculated with eq.
III (Jiang et al., 2021; Bluvshtein et al., 2017).

$$MAC_{Int}(\text{cm}^2 \times g_{TOC}^{-1}) = \frac{\ln(10) \times \int_{250nm}^{600nm} \alpha_\lambda \times d\lambda}{TOC} \times 10^{-6} \ (III)$$

In eq. III, $\alpha_\lambda$ is the base-10 absorbance of the reaction mixture derived by the optical pathway length ($cm^{-1}$), TOC is the concentration of non-purgeable organic carbon ($mg \times L^{-1}$).

The TOC-normalized rate of absorption of sunlight ($R_{abs}$) by the reaction solution was calculated with eq. IV (Jiang
et al., 2021).

$$R_{abs}(\text{photons} \times s^{-1} \times mg_{TOC}^{-1}) = \left( \frac{\ln(10) \times \int_{250nm}^{600nm} \alpha_\lambda \times I_\lambda \times d\lambda}{TOC} \right) \times 10^{-3} \ (IV)$$

In eq. (IV), $I_\lambda$ is the actinic flux ($photons \times s^{-1} \times cm^{-2} \times nm^{-1}$) estimated with TUV calculator for zenith angles 0-50° (Ncar, 2016).

### 2.7 Control experiments and uncertainty

The stability of phenols under investigation in the presence of $H_2O_2$ and UV-Vis irradiation only was studied by carrying control experiments (section S6). Also, for the experiments at pH=2 HCl or $HClO_4$ was used to confirm that the buffering agent used did not affected the distribution of detected products. Control experiments revealed that all phenols under investigation were stable at pH≤7, within the time-scale of the experiments, but 1,2,4-THB, 4-NPG, 5-NPG and HH underwent irreversible reactions at pH>7.

Experimental uncertainties are reported as 2σ from triplicate measurements, other uncertainties were calculated with the exact differential method, unless noted otherwise.



## 3  Results and discussion

### 3.1 Products and reaction mechanism

As presented in Fig. S4, HH, 1,2,4-THB, 4-NC and 5-NPG were formed following reaction (1) under acidic pH

conditions, which is in a good agreement with the previously published results (Xiong et al., 2015; Oturan et al., 2000; Tauber et al., 2000; Liu et al., 2010; Du et al., 2017; Chen et al., 2018). 4-Nitroresorcinol (4-NR) was also tentatively identified among the products. Isomers of 4-NC were previously reported as products of 4-NP oxidation by OH but the exact structures were not proposed for these compounds (Zhao et al., 2013).

To our knowledge, this work is first to report the formation of 4-NPG from reaction (1). (Xiong et al., 2015)

Previously, detection of isomeric products (4-NR and 4-NPG) might have been difficult due to lack of standards and absence of the MS detector (Tauber et al., 2000; Liu et al., 2010; Daneshvar et al., 2007). Also, insufficient resolving power of HPLC used to investigate the composition of products likely contributed to the fact that the formation of 4-NR and 4-NPG was not previously observed (Oturan et al., 2000; Tauber et al., 2000; Liu et al., 2010; Daneshvar et al., 2007; Lipczynska-Kochany, 1991).

The phenolic products from reaction (1) were quantified with GC/MS; the results are presented in Fig. 1.



**Figure 1: Formation of phenolic products following the OH reaction with protonated (A) and deprotonated (B) forms of 4-NP. The slopes were derived from the linear regression analysis of the initial section of the plots, linear coefficients of determination ($R^2$)≥0.97 were obtained. Uncertainties are standard errors from the linear regression analysis.**






Results of the experiments carried out in unbuffered solution are not included in Fig. 1. Due to the formation of nitrite ($NO_2^-$) and nitrate ($NO_3^-$) ions (Kotronarou et al., 1991; Lipczynska-Kochany, 1991; Liu et al., 2010; Kavitha and Palanivelu, 2005), pH of the reaction solution quickly decreased to about 3.5 (Di Paola et al., 2003). Hence, the distribution of products in the acidic and unbuffered solutions was the same.

4-NC was the major product detected (Fig. S4), in the acidic solution, the other products detected were HH, 1,2,4-THB, 4-NPG and 5-NPG. The proposed mechanism of reaction (1) is shown in Fig. 2.

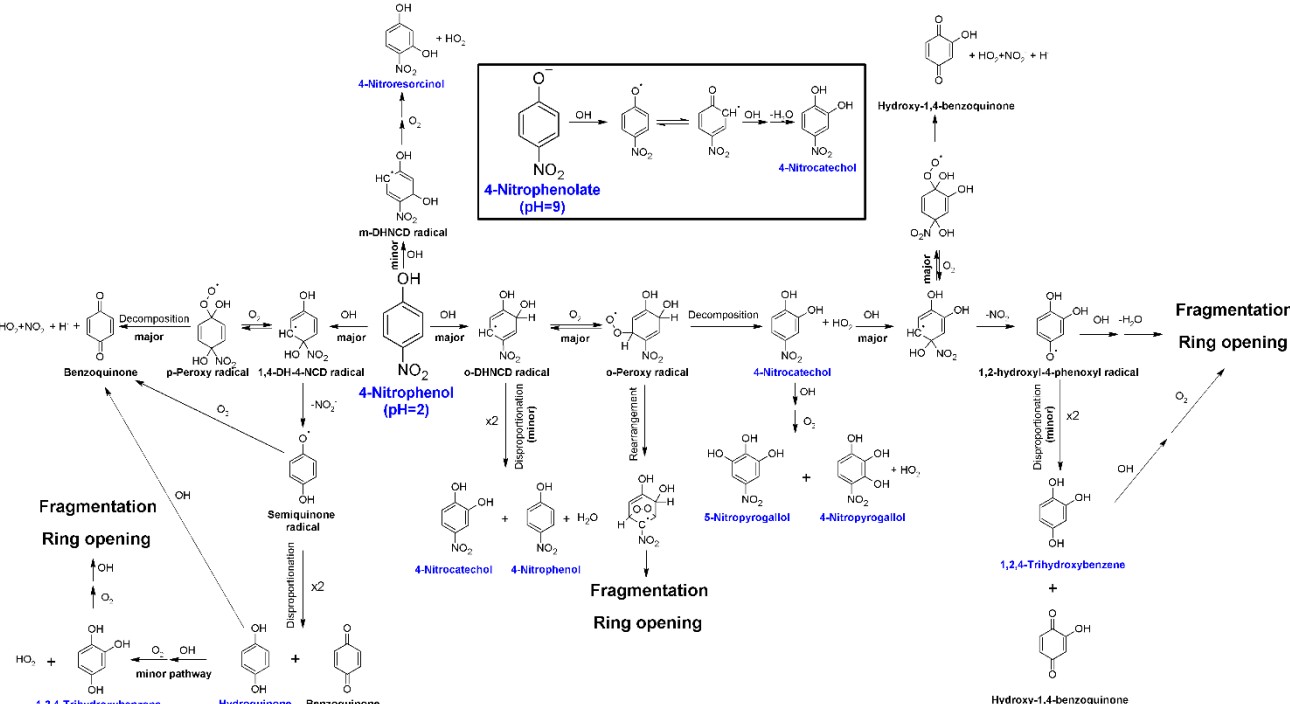

**Figure 2: The proposed mechanism for the reaction of OH with 4-nitrophenol in the aqueous solution. Names of compounds detected in this work are shown in blue.**

As presented in Fig. 2, the electrophilic addition of OH to 4-NP yields dihydroxynitrocyclohexadienyl (DHNCD) radicals (Kotronarou et al., 1991; Di Paola et al., 2003; Kavitha and Palanivelu, 2005). Due to the combined directing effects of -OH (electron-donating) and -$NO_2$ (electron-withdrawing) substituents, OH preferentially adds in the ortho position, resulting in 1,2-dihydroxy-4-nitrocyclohexadienyl radical (ortho-DHNCD radical) (Zhao et al., 2013; Gonzalez et al., 2004; Tauber et al., 2000). Two o-DHNCD radicals can undergo disproportionation reaction to produce 4-NP and 4-NC (Gonzalez

et al., 2004; Tauber et al., 2000; Liu et al., 2010) or react with molecular oxygen (Gonzalez et al., 2004; Di Paola et al., 2003; Oturan et al., 2000). Reaction of o-DHNCD radical with $O_2$ produces the o-peroxy radical which can rearrange, yielding a non-phenolic products, or decompose to produce 4-NC and a hydroperoxyl radical (Di Paola et al., 2003; Kotronarou et al., 1991; Gonzalez et al., 2004; Oturan et al., 2000; Liu et al., 2010). Due to the measured yield of 4-NC of 0.21 (Fig. 1) it is reasonable to assume that the o-peroxyradicals preferentially decompose to ring-opening products. Alternatively, the





decomposition of the o-peroxyradicals yields 4-NC, a major phenolic product detected (Di Paola et al., 2003; Xiong et al.,
        2015; Ding et al., 2016; Oturan et al., 2000; Liu et al., 2010). OH addition in meta position results in the formation of meta-
        DHNCD radical; this minor pathway leads to the formation of 4-NR (Fig. 2). Note that, for clarity, the fragmentation pathways,
        leading to the formation of ring-opening products, are not shown in Fig. 2 for all peroxyradicals.

        4-NPG and 5-NPG are likely formed from 4-NC via analogous mechanism, OH addition to the aromatic ring followed by

reaction with molecular oxygen (Du et al., 2017; Ding et al., 2016; Xiong et al., 2015; Zhang et al., 2003; Oturan et al., 2000).
        This was also confirmed by the experimental data reported in section S7. Interestingly, only trace amounts of 1,2,4-THB were
        formed from OH+4-NC reaction (Fig. S6). When the 4-NC was oxidized in the unbuffered solution (DI water), the initial pH
        was approx. 6.3 but quickly decreased to 3.3 thereby excluding the potential hydrolysis reaction of 1,2,4-THB under these
        experimental conditions (Table S1).

It was previously proposed that 1,2,4-THB is formed by the addition of OH in para position of 4-NC (ipso attack) followed by
        elimination of $NO_2^{\cdot}$ (Daneshvar et al., 2007; Zhang et al., 2003; Kavitha and Palanivelu, 2005), but this assumption was never
        confirmed experimentally. It is still unclear whether or not the 1,4-hydroxy-4-nitrocyclohexadienyl type radicals primarily
        eliminate $NO_2^{\cdot}$ to yield stable phenolic products or decompose to phenoxyl radicals and $NO_2^-$ (O'neill et al., 1978; Kotronarou
        et al., 1991; Xiong et al., 2015; Di Paola et al., 2003; Liu et al., 2010). The limited literature data available indicate that the

elimination of $NO_2^-$ is more likely (O'neill et al., 1978; Kotronarou et al., 1991), which would also explain the fast decrease in
        pH observed during OH+4-NC reaction and the absence of 1,2,4-THB among the major products (Fig. S6). The trace amounts
        of 1,2,4-THB formed from OH+4-NC reaction are likely due to minor disproportionation reaction of the two phenoxy radicals
        (Liu et al., 2010). It was previously reported that 4-NC was quantitatively converted into 1,2,4-THB in the absence of $O_2$,
        which effectively promoted the disproportionation reaction between two 1,2-hydroxyl-4-phenoxyl radicals (Gonzalez et al.,

2004; Liu et al., 2010; Di Paola et al., 2003). These results are in a good agreement with the experimental data acquired in this
        work.

        Ipso addition of OH to 4-NP results in the formation of 1,4-dihydroxy-4-nitrocyclohexadienyl (1,4-DH-4-NCD) radical (Fig.
        2) (O'neill et al., 1978; Daneshvar et al., 2007; Kotronarou et al., 1991). This radical likely eliminates $NO_2^-$ to produce
        benzoquinone (BQ), thereby contributing to the observed decrease in pH in the unbuffered reaction mixture (Al-Suhybani and

Hughes, 1985; Gonzalez et al., 2004; Kotronarou et al., 1991). Consequently, HH is likely produced via disproportionation of
        two semiquinone radicals, which is evidently a more favorable reaction than the analogous reaction between the two 1,2-
        hydroxyl-4-phenoxyl radicals (Kotronarou et al., 1991). Previously, the formation of 1,2,4-THB from the reaction of OH with
        HH was observed (Barzaghi and Herrmann, 2002; Niessen et al., 1988; Sobczyński et al., 2004). Evidently, under the
        experimental conditions used in this work, HH is rapidly oxidized to 1,2,4-THB and BQ resulting in the low yield measured

for this product (Fig. 1A) (Sobczyński et al., 2004; Kotronarou et al., 1991; Di Paola et al., 2003; Gonzalez et al., 2004; Oturan
        et al., 2000).

        Lower number of products was observed at pH=9, likely due to hydrolysis or other reactions of HH, 1,2,4-THB, 4-NPG and
        5-NPG. Additionally, the yield of 4-NC was increased (Fig. 1B) at pH=9. The 4-nitrophenolate (Fig. S1) is expected to react





via mixed mechanism: one-electron oxidation and OH addition - Fig. 2 (Biswal et al., 2013; Zhao et al., 2013). The subsequent
reactions of 4-nitrophenolxyl radicals formed following the one-electron oxidation are unclear (Gonzalez et al., 2004; Liu et
al., 2010; Zhao et al., 2013; Wojnárovits and Takács, 2008). Similar, semiquinone radicals were shown to react with $O_2$ or
disproportionate (Valgimigli et al., 2008; Gonzalez et al., 2004). However, these reactions cannot explain the much higher
yield of 4-NC in basic solution (Fig 1B). It was proposed that the resonance forms of phenoxyl radicals may react directly with
OH  and $NO_2$, which leads to the formation of stable phenolic products (Fig. 2) (Barzaghi and Herrmann, 2002; Niessen et al.,
1988). This mechanism would explain factor of 2 higher yield of 4-NC in basic solution (Fig. 1B). Analogous reaction of 1,2-
hydroxyl-4-phenoxyl radical likely leads to ring-opening products, as previously reported (Liu et al., 2010).

In a concentrated (160 mM) basic solutions, 1,2,4-THB  was shown to generate stable aromatic oligomers with the absorbance
between 400-700 nm (Randolph et al., 2018). Detecting such oligomers with GC/MS is unlikely due to their lower volatility,
insufficient thermal stability or low reactivity towards AA (section 2.3). However, at pH>7 the integrated absorbance of the
reaction solution in this spectral range is lower as compared with the acidic solution, as discussed in more detail in section 3.2.
Hence, the formation of "brown" phenolic oligomers from 1,2,4-THB is evidently suppressed in a more diluted solution and
in the presence of nitrated phenols, $NO_2^-$ and $NO_3^-$ ions.

## 3.2 Light absorptivity and the time-evolution of brown-carbon chromophores

A continuous decrease of the absorbance of the reaction solution was observed (section S8). On the other hand, an initial, small
increase in the absorbance at 420 nm of the 4-NP solution during reaction with OH was previously reported followed by a
rapid bleaching (Zhao et al., 2015); such differences can be caused by slightly different reaction conditions. Also, in this work,
an integrated absorbance values were used (eq. I and II) which may be a more adequate approach due to shifting of the
absorption maximum ($A_{max}$) of the reaction solution (Fig. S8) (Zhao et al., 2015; Hems and Abbatt, 2018).

The contribution of the light-absorbing products of reaction (1) to the overall light absorptivity of the reaction solution was
evaluated via eq. I. Results presented in Fig. S10 show that when the reaction was carried out under basic pH conditions the
relative absorbance of products (eq. I) was lower and increased  slowly. This points out that the light-absorbing compounds
are not stable at pH>7 which is in a good agreement with the results discussed in section 3.1 (Randolph et al., 2018).

The values of $MAC_{init}$ and $R_{abs}$ calculated with eq. III and IV are presented in Fig. 3.





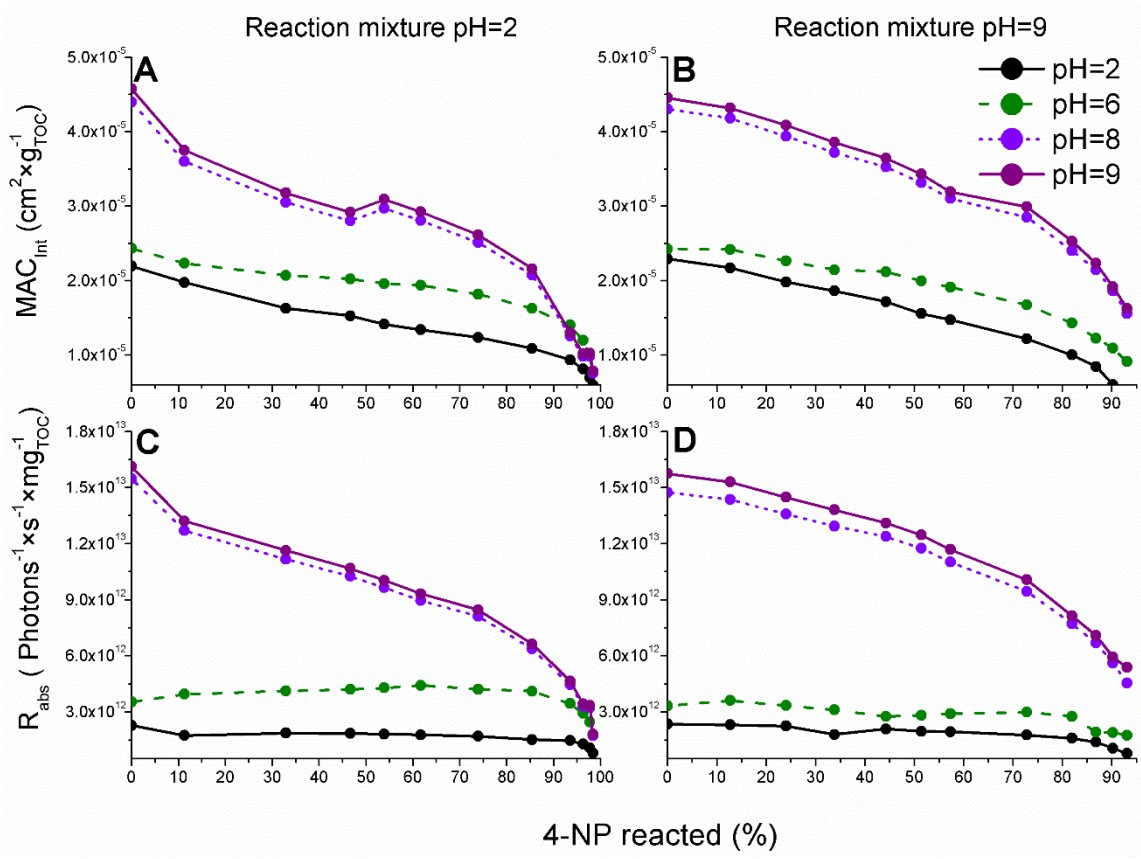

**Figure 3:** The pH dependent organic carbon-based mass absorption coefficients (MACint) derived using the integrated absorbance peak for the reaction mixture absorptivity measured for the reaction of 4-NP(A) and 4-nitrophenolate (B) and the corresponding TOC-normalized rates of sunlight absorption (Rabs) for of 4-NP (C) and 4-nitrophenolate (D). Only data for pH =2,6,8, 9 is shown for clarity and complete data is presented in Fig. S12. Experimental data is represented by points, lines are provided to guide the eye.

As expected, MACint decreased steadily following the oxidation of the precursor. The experimental data acquired (Fig. 3) show a clear increase in absorptivity following the increase in pH at which the absorbance was measured due to pH-depended $\varepsilon$ values for the light-absorbing phenols present in the reaction solution. In Fig. 3A and B, the disappearance rates of MACint are of similar order (Table S3). This is most likely due to the formation of higher number of light-absorbing phenols (second-generation products) at pH=2 and due to higher yield of 4-NC (which is characterized by high $\varepsilon$ values, Fig. S5) at pH=9, respectively. Consequently, the rates of disappearance of the overall light absorptivity of 4-NC following reaction (1) are mostly independent on pH of reaction solution and primarily depend on pH at which the absorbance is measured. The MACInt values calculated were slightly higher as compared with the values measured for the previously investigated aromatic BrC chromophores (for non-nitrated precursors) (Jiang et al., 2019; Jiang et al., 2021), likely due to the high $\varepsilon$ values of 4-NP and nitrated phenols formed following reaction (1).





The $R_{abs}$ values (Fig. 3C and D) decrease more slowly as compared with the values of $MAC_{Int}$, which is caused by a red-shift

of $A_{max}$ of the reaction solution combined with the pH – dependence of $\varepsilon$ values of the 4-NP and phenolic products (for instance,

red-shift of $A_{max}$ following increase of pH observed for 4-NC, Fig. S5). Also, because of a significant increase in the actinic

flux at $\lambda > 400$ nm (Fig. S14), any "brown" products formed efficiently stabilize the $R_{abs}$ values thought the course of the reaction

when pH<7 - see also Fig. S12.

**4. Conclusions**

The average measured Henry's law constant, $5 \times 10^4$ ($M \times atm^{-1}$), indicates that 4-NP resides entirely in the aqueous-phase in

clouds but not in "wet" aerosols (Fig. S13) where it can undergo chemical and photochemical processing (Herrmann et al.,

2015).

The first-order $k_{bleaching}$ coefficients derived via eq. II (Table S3) show that the lifetimes of  BrC chromophores are 3 and 1.5-

times longer than the lifetime of 4-NP (precursor) under acidic and basic pH conditions, respectively, due to formation of light-

absorbing products. The rates of bleaching of 4-NP solution due to reaction with OH and due to direct photolysis were

compared (Fig. 4) using the $k_{bleaching}$ values derived in this work and the previously reported, average quantum yields ($\phi$,

molecules×photon$^{-1}$) – see Table S3 and eq. SIII and IV (Braman et al., 2020; Lemaire et al., 1985; Biswal et al., 2013; García

Einschlag et al., 2003). The $\phi$ values listed in Table S3 were derived by measuring decrease in the absorbance of 4-NP solution,

hence can be regarded as effective $\phi$ for the bleaching of 4-NP and 4-nitrophenolate-derived BrC. To our knowledge, the

wavelength-dependent $\phi$ values for 4-NP and 4-nitrophenolate are not available.

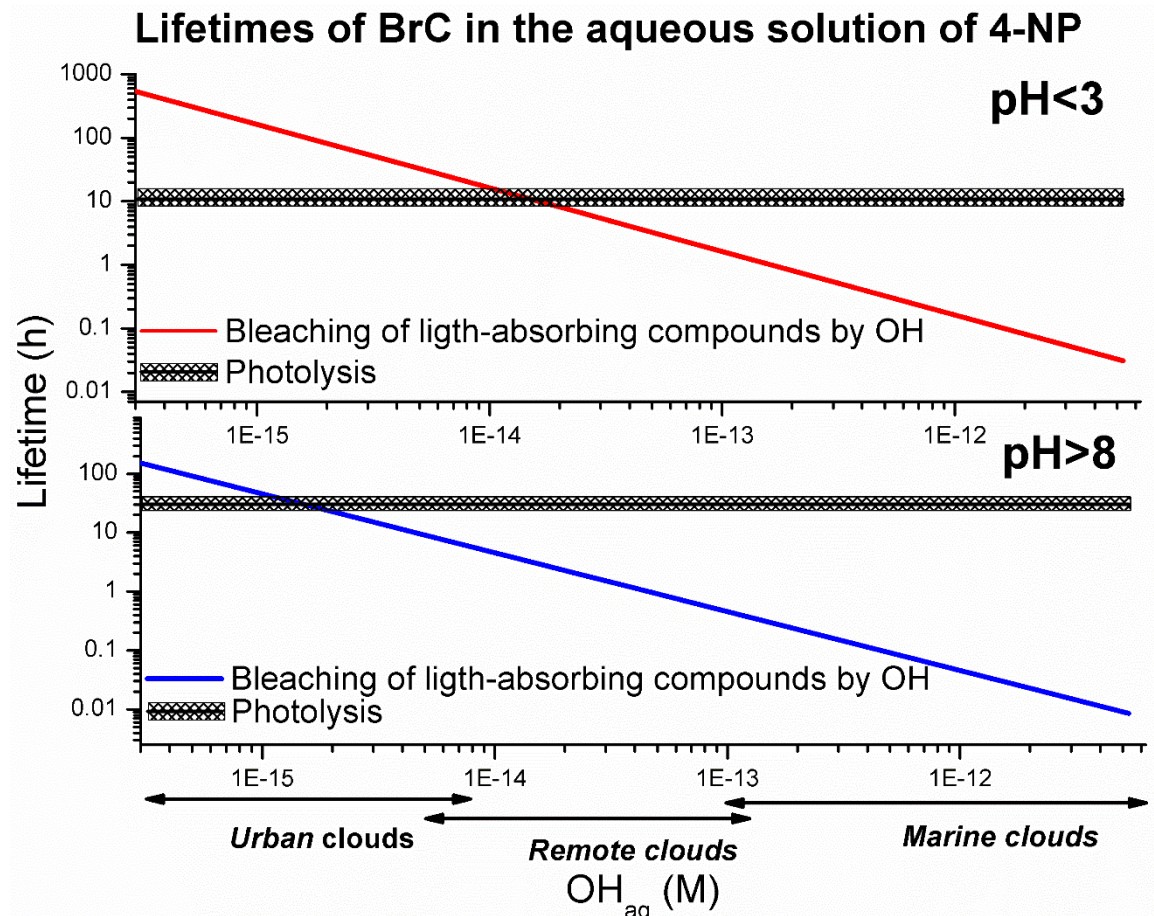

**Figure 4: The estimated aqueous-phase lifetimes of light-absorbing compounds in the solutions of 4-NP and 4-nitrophenolate due to reaction with OH and direct photolysis. The lifetimes due to reaction with OH were calculated with kbleaching coefficients derived via eq. II with the data acquired in this work. The average lifetime due to photolysis is shown for zenith angles 0-50°, shaded area is 2σ, representing the range of photolysis lifetimes calculated via eq. SIV.**

As presented in Fig. 4, both bleaching mechanisms can be relevant under realistic atmospheric conditions, depending from [OH]. Bleaching by OH is expected to be a more dominant pathway for 4-nitrophenolate, due to its lower reported quantum yields ($\phi$) combined with higher OH reactivity of the precursor at pH>8 (Lemaire et al., 1985; Braman et al., 2020).

Previously, the reaction with the OH was concluded to be the major removal mechanisms for a number nitrophenols in the atmospheric aqueous phase (Zhao et al., 2015; Vione et al., 2009; Albinet et al., 2010). In the context of the formation and processing of the atmospheric BrC, reaction of OH with 4-NP leads to the removal of light-absorbing compounds. At the same time, it was previously concluded that reaction of OH with 5-nitroguaiacol, 4-NC and dinitrophenol initially lead to the increase in the light-absorptivity followed by a rapid bleaching of the reaction solution (Hems and Abbatt, 2018; Zhao et al., 2015).

The results described in this work and the literature data (Hems and Abbatt, 2018; Zhao et al., 2015) indicate that more substituted nitrophenols initially yield higher amounts of light-absorbing products as compared with 4-NP. Moreover, lifetimes



of BrC chromophores are expected to be significantly longer than the lifetimes of the parent nitrophenols (precursors) due to the formation of aromatic, light-absorbing by-products (Hems and Abbatt, 2018; Zhao et al., 2015).

Based on GC/MS quantitative data it was estimated that in reaction (1) ca. 20 to 40% of 4-NP, depending on pH, was converted

into phenolic products. A low degree of mineralization of the precursor (ca. 15% - section S15) and up to 40%   yield of phenolic products indicates that reaction (1) generates a substantial amount, between 45 to 65%, of non-aromatic products, like for instance functionalized carboxylic acid (Hems and Abbatt, 2018; Kavitha and Palanivelu, 2005; Zhang et al., 2003; Oturan et al., 2000). Consequently, reaction (1) and reaction of other nitrophenols with OH can contribute to acidity of atmospheric aqueous particles via formation of $NO_2^-$ , $NO_3^-$ and organic (nitrated) acids (Tilgner et al., 2021). Additionally,

aqueous oxidation of nitrophenols via OH may be a source of potentially toxic and harmful aqueous SOAs ($_{aq}$SOAs).

*Data availability.* The raw data can be obtained by contacting the corresponding author.

*Author contributions.* BW designed the study, developed the methodology analyzed the data and wrote the paper PJ carried out the experiments, optimized the methodology and processed the raw data, TG supervised the experiments, analyzed the data and contributed to the final manuscript. All authors contributed to the interpretation of the results.

*Competing interests.* The authors declare that they have no conflict of interest

*Acknowledgments.* This work was carried out at the Biological and Chemical Research Centre, University of Warsaw, established within the project co-financed by European Union from the European Regional Development Fund under the Operational Programme Innovative Economy, 2007 – 2013. We thank dr Bartłomiej Kiersztyn for making the TOC measurements possible. We thank dr Marcin Wilczek for the NMR measurements.

*Financial support.* This project was founded funded by the Polish National Science Centre: grant number UMO-2018/31/B/ST10/01865. The authors also acknowledge support from the statutory research funds of the University of Warsaw, decision number BOB-661-199/2020.

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
