# Peer review of "Aqueous chemical bleaching of 4-nitrophenol brown carbon by hydroxyl radicals; products, mechanism and light absorption"

_Atmospheric Chemistry and Physics, 2021_

## Referee Comment (RC1)

**General comments**

The paper presents the results on aqueous-phase reactions of 4-nitrophenol (4NP) with OH radicals leading to new products formation. Depending on pH (2 or 9), about 20 to 40% of 4NP was converted into new aromatic light-absorbing compounds, with the highest contribution of 4-nitrocatechol (4NC). Besides, up to 65% of organic carbon in the reaction solution (after 4NP was completely consumed) represented the non-aromatic open-ring compounds. Consequently, the light absorption of the solution decreased with time (i.e., bleaching of the reaction solution), however with some prolongation due to initially formed aromatic compounds.

There are many open questions concerning mechanisms of brown carbon (BrC) formation, especially those in cloud droplets and aqueous particles. But, more and more studies confirmed the importance of aqueous-phase (photo)chemical processing in contribution to organic aerosol aging, and so to light-absorbing secondary aerosol formation/degradation. The topic is certainly actual.

However, the manuscript is written superficially, it is sometimes confused and not well readable, sometimes due to not precise expressions, not good choice of words or due to grammatical errors. Besides, there is too much material, too many results in the Supplement, which needs to be checked frequently to follow the results and discussion in the manuscript.

Conditionally, the manuscript could be of adequate atmospheric interest to merit publication in *Atmospheric Chemistry and Physics*, but after major revision, with addressing the following comments and/or questions. Besides, I highly recommend the English language checking, some parts should be re-written. Below, I list only very few language-related errors in the main text.

**Specific comments**

Introduction:
- Line 25: The authors may add a reference of Hems et al., ACS Earth Space Chem. 2021.
- Line 28: The authors may add a reference of Vidović et al., Atmosphere, 2020.
- Line 32: I suggest to include also the references for example: Claeys et al., Environ. Chem., 2012; Kitanovski et al., J. Chromatogr. A 2012; Frka et al., Environ. Sci. Technol., 2016.
- Line 35: The statement is not entirely true; there are other nitroaromatic compounds (NAC), which are even more important BrC components (e.g., 4-nitrocatechol, 4-NC; etc.). Xie et al., 2019 (this ref. is cited), demonstrated that among 14 NACs identified in biomass burning (BB) samples and also in simulated SOA, 4-NC contributed the most to overall BrC absorption at 365 nm (see Fig. 4. in Xie et al., 2019).
- Line 39: Here, the references Kitanovski et al., J. Chromatogr. A 2012; Claeys et al., Environ. Chem., 2012; Frka et al., Environ. Sci. Technol., 2016 should be included as well.
- Line 48: The authors could add a reference of Hems et al., ACS Earth Space Chem. 2018.
- Lines 55-57: It would be better as: "...the connection between the light absorption and chemical composition..."
- Reaction is usually written as: 4NP + OH $\rightarrow$ (check throughout the text)

- Line 71: This is not entirely true. Atmospheric aqueous particles have usually low pH (depends on their origin, but mostly below 3), while other atmospheric liquid waters (e.g., cloud droplets, fog) have mostly higher pH values (above 3); see Table 1 in Herrmann et al., 2015.

Experimental:
- Chemicals should be involved.
- Although the reactor is described in the Supplemental, I strongly suggest describing it at least briefly in the manuscript.
- Line 93: As explained in S4.1, in addition to two UVC lamps (for the photolysis of $H_2O_2$) also six lamps (Vis above 400 nm) were used.
- Line 96: Deionized $H_2O$ is not good enough for such kind of experiments; usually high purity water should be used.
- Line 101 and 2.3.: Why did you use GC-MS? Wouldn't be easier and faster by LC-MS (no derivatization)?
- Line 118: Why adjusted again before UV-Vis measurements (you did this at the beginning of experiment)? In this way, you did not have the same conditions as in the reaction solution.
- Lines 116-121: Very awkwardly written, and thus unclear. From the text in the main manuscript, it should be clear how the measurements were done (the supplemental material should provide only the additional and more detailed information).
- Line 123: Non-purgeable organic carbon: What do you mean by non-purgeable OC?
- In Eq. II, change $K_{A,rmix}$ with $k_{A,rmix}$ as it is written in line 141; the same for $K_A$, (first-order rate constants).
- Line 140: Instead of " bimolecular reaction rate coefficient", "second-order rate constant" should be used. Please, check throughout the manuscript and SI.
- Line 141: …first-order disappearance rate constants…. ?
- Line 147: Add $d_\lambda$ (absorbing path length, it is in cm and not in $cm^{-1}$). I also suggest using the same characters for the same parameters as usually used for MAC (Laskin et al., Chem. Rev. 2015).
- Line 157: HCl and $HClO_4$ are acids (not buffers)!

Results and discussion
- Too much material in Supplement, more should be reasonably involved in the manuscript.
- Fig. S4 should be involved in the main MS.
- Line 167/168: Which isomers of 4NC do you have in mind?

- Fig.1: What does it present: the dependence of conc. of products vs. conc. of initial 4NP? One can conclude that with a higher initial concentration of 4NP, higher conc. of 4NC was formed (at pH 2, 3 other products as well), but only to a certain extent. Can you give some explanation?

- Fig. 1: Especially in the case at pH 9, it is not correct to derive the slope from a linear regression analysis.

- Lines 97, 181/184, etc.: "unbuffered" solution: Do you mean that the reaction solution was not adjusted to a certain pH using buffer (or only not adjusted)? However, as it can be seen you did measure the initial pH of such reaction solution (in SI, Fig. S6).

- Line 184: As I understand, the authors concluded that the distribution of products was the same in both cases (in aqueous solution with pH 2 and in that with unadjusted initial pH). I assume that your conclusion is based on fact that in both cases the reaction mixtures were acidic at the end. The authors have to be more precise in the formulation to clarify the text.

- Lines 185-245: Since the whole part is confused, I recommend shortening and writing the text more concisely explaining the mechanism with emphasize on the main formation pathways (shown in Fig. 2), and on your findings.

- Line 282: …"where it can undergo chemical and photochemical processing": What this statement refers to, clouds or wet aerosol, or both? From what has been written, one would conclude that the processes take place only in wet aerosols.

- Line 297: Which two bleaching mechanisms: via OH reactions and via photolysis? From the results in Fig. 4, photolysis is not very effective.

**Technical corrections**
- All references (in parentheses) have to be written from the earliest to the latest one according to the year of publication.
- I suggest changing "absorptivity" with "absorption": in the title and throughout the manuscript: e.g., line 26: it should be "UV-Vis absorption"; line 54: "light absorption of aqueous particles", etc.
- Line 54: The chemical and photochemical…..result (not results).
- Line 76: Should be plural (…are strongly..).
- Line 84: Should be plural (…were monitored..).
- Line 90: Aqueous-phase reactor (here "aqueous-phase" is an adjective)
- Line 105: Delete ", the instrument was"; it should be "and equipped with…"
- Line 123: "was quantified" (or determined)
- 4-nitophenol (4-NP) can be written as 4NP, etc.
- Base-e, base-10: it is no need to write all the time; it's obvious from the equations.
- Fig.3: Data are presented…(plural)
- Line 281: Instead of "resides" it's better "exists"
- Line 297: …depending on [OH]
- Page 15: Authors of the first reference are missing.

**Supplemental material**

P. 3, line 33: Not "allowed", but "used"
Table S1: Give the concentration ranges in mg $L^{-1}$.

---

## Author Comment (AC1)

**Biuro Tłumaczeń UZUS**
**ul. Winogrady 120 A**
**61-626 Poznań**
**http://www.uzus.pl**
e-mail: **biuro@uzus.pl**
tel. (+48) 606 630 989

I hereby declare that the article entitled 'Aqueous chemical bleaching of 4-nitrophenol brown carbon by hydroxyl radicals; products, mechanism and light absorption' by Bartłomiej Witkowski, Priyanka Jain and Tomasz Gierczak has been proofread by a native speaker of English, a professional proofreader of academic articles.

Yours sincerely,

*Barbara Komorowska*

Translation Agency UZUS

---

## Author Response (AR1)

We would like to thank the anonymous reviewers for their very insightful comments and suggestions. Our point-by-point responses to the specific comments are provided below. The revised manuscript was also proofread and corrected by a professional English proofreading service (see the enclosed certificate of correction). In a few cases, we combined referees comments that referred to the same portions of the manuscript, most notably the comments about Fig. 2 (formerly Fig. 1).

**Referee #1**

-There is too much material, too many results in the Supplement, which needs to be checked frequently to follow the results and discussion in the manuscript.

**Author's response:** We agree, the manuscript submitted indeed reports a substantial amount of experimental data. At the same time, we believe that publishing this material in two (or more) separate articles was not a feasible option. Consequently, while preparing the initial and revised manuscripts, we were focused on maintaining a reasonable balance between the volumes of the main text and supplement and experimental details that are included in the main text.

**Changes in the manuscript:** The specific changes made in the revised manuscript are listed below. We did our best to adequately address all comments kindly provided by the referees while maintaining a reasonable volume of the revised manuscript. In the revised manuscript, Fig. S4, S11, S15, and Table S3 were all moved to the main text; the discussion connected with these figures and with Table S3 (now Table 1) was appropriately revised. At the same time, we believe that including too many experimental details and results in the main text would lead to the decrease in the readability of the manuscript.

**Responses to the specific comments and changes made in the revised manuscript:**

Line 25: The authors may add a reference of Hems et al., ACS Earth Space Chem. 2021.

Line 28: The authors may add a reference of Vidović et al., Atmosphere, 2020.

Line 32: I suggest to include also the references for example: Claeys et al., Environ. Chem., 2012; Kitanovski et al., J. Chromatogr. A 2012; Frka et al., Environ. Sci. Technol., 2016.

**Author's response:** These references were added to the introduction in the sentences listed by the referee.

**Line 35:** The statement is not entirely true; there are other nitroaromatic compounds (NAC), which are even more important BrC components (e.g., 4-nitrocatechol, 4-NC; etc.). Xie et al., 2019 (this ref. is cited), demonstrated that among 14 NACs identified in biomass burning (BB) samples and also in simulated SOA, 4-NC contributed the most to overall BrC absorption at 365 nm (see Fig. 4. in Xie et al., 2019).

**Author's response:** 4NP was confirmed as one of the most abundant nitro-monoaromatic hydrocarbons in the ambient PM(Kitanovski et al., 2020; Bluvshtein et al., 2017), frequently with concentration comparable to 4-NC. Thus, the statement that "4NP is one of the major BrC chromophores" is supported by the cited literature.

**Changes in the revised manuscript:** This statement was softened in the revised manuscript "For these reasons, 4NP was found to contribute significantly to the light absorption of ambient BrC aerosols.". Cited references were reviewed and revised appropriately.

**Line 39:** Here, the references Kitanovski et al., J. Chromatogr. A 2012; Claeys et al., Environ. Chem., 2012; Frka et al., Environ. Sci. Technol., 2016 should be included as well.

**Author's response:** The listed references were included in this line, with the exception of Frka et al., Environ. Sci. Technol., 2016; this article did not report the detection of 4NP in the ambient PM.

**Line 48:** The authors could add a reference of Hems et al., ACS Earth Space Chem. 2018.

**Author's response:** The reference listed (Hems and Abbatt, 2018) was included in this sentence.

**Lines 55-57:** It would be better as: "…the connection between the light absorption and chemical composition…"

**Author's response:** This sentence was revised as suggested by the referee.

Reaction is usually written as: 4NP + OH (check throughout the text)

**Author's response:** In the main text as well as in the SI, 4-nitrophenol is abbreviated as 4NP instead of 4-NP and 4-nitrophenolate is now abbreviated as 4NPT

**Line 71:** This is not entirely true. Atmospheric aqueous particles have usually low pH (depends on their origin, but mostly below 3), while other atmospheric liquid waters (e.g., cloud droplets, fog) have mostly higher pH values (above 3); see Table 1 in Herrmann et al., 2015.

**Author's response:** The referee is correct, the pH of ambient aerosols (other than clouds and fogs) is primarily acidic. At the same time, LWC of such aerosols is too low for the 4NP

($H \approx 5 \times 10^{-5}$ M×atm$^{-1}$) to efficiently partition into these water-containing particles. In the pH range of other particles listed in the mentioned review article (Table 1) with sufficiently high LWC (clouds and fogs), 4NP can exist in both protonated and deprotonated forms, due to their higher pH values. Close to the equivalence point (pH=pKa), which is the upper limit of the estimated pH for some clouds and fogs, both forms of 4NP exit in the 1:1 molar ratio (pKa≈7.2 – see also Fig. S1).

**Changes in the revised manuscript:** This sentence was revised as: "In some clouds and fogs (Herrmann et al., 2015), 4NP (pKa≈7.2) (Rived et al., 1998) can exist in both protonated and deprotonated forms (Fig. S1)."

Chemicals should be involved (in main text)

**Author's response:** We believe that listing standards, reagents and materials that were all obtained from commercial suppliers in the main text will not enhance the scientific value of the article. It would also make the revises article somewhat less readable due to adding a substantial amount of volume in to the experimental section.

Although the reactor is described in the Supplemental, I strongly suggest describing it at least briefly in the manuscript.

**Author's response:** We believe that referring the reader to our previous work (Witkowski et al., 2019) together with the description of this setup in section S4.1. provides sufficiently detailed description of this experimental setup. Of course, we will be happy to share the details of our design with any researcher that will contact us with inquiries of questions about this setup.

**Line 93:** As explained in S4.1, in addition to two UVC lamps (for the photolysis of $H_2O_2$) also six lamps (Vis above 400 nm) were used.

**Author's response:** Line 93 was changed accordingly to referee suggestion, P/N of the lamps used in the photoreactor were added in section 2.1.

**Line 96:** Deionized $H_2O$ is not good enough for such kind of experiments; usually high purity water should be used.

**Author's response:** In our experiments, we used the freshly prepared, highest-purity water available on-site in a relatively large quantities. DI water (18 MΩ ×cm$^{-1}$) was previously used in a number of studies to investigate atmospherically-relevant aqueous reactions – see for instance (Aljawhary et al., 2016; He et al., 2019; Harrison et al., 2020; Richards-Henderson et al., 2014) Therefore, we believe that using DI water is adequate.

**Changes in the revised manuscript:** The resistance of Milli-Q water used in all experiments ($18\ M\Omega \times cm^{-1}$) was added at the beginning of experimental section: "All solutions were prepared using deionized (DI) water ($18\ M\Omega \times cm^{-1}$)." Further mentions of DI water after this point were removed to avoid unnecessary repetitions.

**Line 101 and 2.3.:** Why did you use GC-MS? Wouldn't be easier and faster by LC-MS (no derivatization)?

**Author's response:** Naturally, both hyphenated techniques (GC/MS and LC/MS) possess certain advantages and disadvantages. In the case of the submitted work, the use of GC/MS was adequate, as confirmed by the results of the instrument calibration with pure standard - Table S1. Note that sample preparation is also needed prior to the LC/MS analysis, including neutralization of $H_2O_2$, removal of particles from each sample injected into the instrument with single-use syringes and syringe filters (increased costs) and neutralization of strongly acidic and basic samples to protect the LC column– see for instance the procedure described in our recent work (Witkowski et al., 2021). Also, due to very high resolving power of capillary GC and, also due to existence of the searchable mass spectra libraries in GC/MS, we were able to identify 4-nitroresorcinol and 4-nitropyrogallol as products of the investigated reaction.

**Lines 116-121:** Very awkwardly written, and thus unclear. From the text in the main manuscript, it should be clear how the measurements were done (the supplemental material should provide only the additional and more detailed information).

**Line 118:** Why adjusted again before UV-Vis measurements (you did this at the beginning of experiment)? In this way, you did not have the same conditions as in the reaction solution.

**Author's response:** We agree, this procedure was not well explained in the listed paragraphs. The sample handling before the absorbance measurement was as follows. If the pH of the reaction solution was acidic (pH=2) the absorbance of each aliquot was firstly measured at this pH, then it was adjusted by 1 pH unit and it was measured again. This was repeated for each aliquot of the reaction solution until pH=9 was reached. Analogous procedure was used to measure the absorbance of the aliquots of the reaction solution when the reaction was carried out under basic conditions (pH=9), but the pH was adjusted by 1 pH unit starting from 9 and finishing at 2. This procedure also outlined in more detail in the SI section S4.3.

**Changes in the revised manuscript:** Section 2.4 was revised to clarify the sample preparation procedure for the UV-Vis measurements.

**Line 123:** Non-purgeable organic carbon: What do you mean by non-purgeable OC?

**Author's response:** We were hoping that the term NPOC would be self-explanatory to most readers of ACP, since it is frequently used in the studies utilizing TOC instruments. In TOC instruments, the samples are frequently sparged with oxygen to remove the dissolved $CO_2$; in this process, VOCs that are sparingly soluble in water are also removed. Therefore, NPOC refers to the fraction of organic carbon that is not removed following the sparging with $O_2$.

**Changes in the revised manuscript:** Section S4.4 was added in the SI, providing a more detailed description of the TOC measurements. The description of TOC measurements in section 2.5 was revised slightly.

**Line 140:** Instead of " bimolecular reaction rate coefficient", "second-order rate constant" should be used. Please, check throughout the manuscript and SI.

**Line 141:** …first-order disappearance rate constants…. ?

**Line 147:** Add d$\lambda$ (absorbing path length, it is in cm and not in $cm^{-1}$). I also suggest using the same characters for the same parameters as usually used for MAC (Laskin et al., Chem. Rev. 2015).

**Author's response:** These changes were incorporated accordingly to the referee's suggestions, the listed reference was added in section 2.6.

**Line 157:** HCl and $HClO_4$ are acids (not buffers)!

**Author's response:** The referee is obviously correct, these inorganic compounds are acids, not buffers. However, in the manuscript they are referred to as "buffering agents" which, to our knowledge, is a term used to describe agents/compounds added to the aqueous solutions to adjust the pH. In the presented study, these two inorganic acids were used to adjust the pH of the reaction solution, hence we used the term buffering agents.

**Results and discussion**

Too much material in Supplement, more should be reasonably involved in the manuscript.

**Author's response:** While incorporating more results in the revised article, we did our best to maintain a reasonable volume of the revised text. Changes made in the revised manuscript are listed below, together with our responses to the specific referee's comments.

**Fig. S4** should be involved in the main MS.

**Author's response:** We agree with this suggestion.

**Changes in the revised manuscript:** Fig. S4 was moved to the main text – now Fig. 1.

**Line 167/168:** Which isomers of 4NC do you have in mind?

**Author's response:** (Zhao et al., 2013) et al. observed three chromatographic peaks using LC/MS in the negative ionization mode from the 4NP+OH reaction in the presence of oxygen - see also Table 1 in the supplement of this article. However, no structures were assigned to the two isomers of 4NC detected, these products were also not included in the reaction mechanisms proposed by the authors.

**Changes in the revised manuscript:** This sentence was revised as "Previously, two unknown isomers of 4NC were detected as products of reaction (I) (Zhao et al., 2013)".

**Referee 1:**

**Fig.1:** What does it present: the dependence of conc. of products vs. conc. of initial 4NP? One can conclude that with a higher initial concentration of 4NP, higher conc. of 4NC was formed (at pH 2, 3 other products as well), but only to a certain extent. Can you give some explanation?

**Referee 3:**

**In Figure 1A and 1B,** what is the x-axis? Is this the change in 4NP concentration relative to the initial concentration? Please add this to the caption.

**Author's response:** We agree that the process of deriving molar yields of the products formed was not well explained in the original manuscript.

The x axis in Fig. 2 (formerly Fig. 1), represents the amount of 4NP consumed during the course of the reaction (labeled as $\Delta$ 4-Nitrophenol, mM). The y axis in Fig. 2 is the concentration of the individual product. Both plots (Fig. 2A and Fig. 2B) start from the point 0,0 because, at the onset of the reaction ($t_0$), the amount of 4NP consumed by the OH is 0 (mM), the same as the concentration of the quantified products.

The yield for a given product is derived from the slope of the initial, linear section of these plots, which is a well-established approach when analyzing chemical reactions – see for instance (Gierczak et al., 2021; Chattopadhyay et al., 2021). Therefore, a higher value of the slope indicates a more efficient conversion of the precursor into a given product.

Note that, due to the secondary chemistry that occurs at the larger stages of the reaction, in this case, OH reaction with the products, the yields (slopes) are derived from the initial (linear) sections of the plots. If the reaction is carried out until the precursor (4NP) becomes almost completely consumed, the expected curving of the plots derived via eq. (I) is observed (Gierczak et al., 2021; Chattopadhyay et al., 2021).

**Changes in the revised manuscript:** Sections 2.3 were and S4.2. were renamed "Quantification of the phenolic products with gas chromatography coupled to mass spectrometry". Eq. I was added in section 2.3, describing how the plots shown in Fig. 2A and 2B were derived. Reference (Gierczak et al., 2021) was added in section 2.3 to underline that the described method for deriving molar yields is adequate.

**Referee 1:**

**Fig. 1**: Especially in case of pH 9, it is not correct to derive the slope from a linear regression analysis.

**Referee 3:**

**In Figure 1A and 1B**, what are the linear fits showing? The data does not look linear, especially in Figure 1B, so why is this assumption being made?

**Author's response:** To evaluate whether or not the applied linear regression model is adequate, p-values and values of the standardized residuals obtained following the linear regression analysis of the data presented in Fig. 2A and 2B were examined. In both cases, p-values $< 0.05$ were obtained, confirming that the linear correlation was statistically significant. Moreover, all values of standardized residuals were $< 3$, confirming that none of the data points included in the linear regression analysis should be classified as an outlier.

**Changes in the revised manuscript:** p-values were added in Fig. 2A and 2B, Table S2 was added in the SI, listing the raw data presented in Fig. 2, residuals and standardized residuals obtained following the linear regression analysis of this data. Table S2 is referenced in the caption of Fig. 2.

**Referee 2:**

Redo Figure 1, it is not very clear. Separate legend from graph (you can put this into the caption).

**Author's response:** As outlined above, the following commends and suggestions kindly provided by referees 1 and 3, Fig. 1 (now Fig. 2) was appropriately revised. At the same time, we believe that separating the legend from the plot, particularly placing the color-codes for the compounds under investigation in the caption, would decrease the overall readability of this figure.

**Changes in the revised manuscript:** Abbreviations used in the main text and in the SI for the aromatic products quantified were added in the revised Fig. 2.

**Referee 3:**

**In figure 1A**, what is the inset? Please add labels to the axis and discuss this figure in

the caption and in the text.

**Author's response:** We agree, the inset decreased the overall readability of the figure and it was not discussed in the manuscript and it is not needed to discuss the yield of aromatic products quantified.

**Changes in the revised manuscript:** The inset was removed; we believe that it was not necessary to adequately present and discuss the results shown in Fig. 2 (formerly Fig. 1). We believe that the revised Fig. 2 is now more readable.

**Lines 97, 181/184**, etc.: "unbuffered" solution: Do you mean that the reaction solution was not adjusted to a certain pH using buffer (or only not adjusted)? However, as it can be seen you did measure the initial pH of such reaction solution (in SI, Fig. S6).

**Author's response:** Unbuffered reaction solution refers to an aqueous solution of the precursor (4NP or 4NC) in DI water with the added $H_2O_2$ and without any buffering agents. The initial pH of these solutions can be measured without adding any acids or bases and they were close to neutral, as expected. Due to the formation of inorganic acids (likely $HNO_3$ and $HNO_2$), the pH quickly decreased to <3, as discussed in the paragraph below Fig. 2 (formerly Fig. 1).

**Changes in the revised manuscript:** Sentence in line 97 was revised as "The pH of this solution was not adjusted (unbuffered, no acids or buffers added) or it was adjusted to pH 2 or 9 with HCl, $HClO_4$ or $Na_2HPO_4$ (50 mM) to investigate the reaction of fully protonated and deprotonated forms of 4NP (Fig. S1)" is used without any additional clarification. Analogously, this explanation (no acids or buffers added) was added in section S4.1. in the SI and the term "unbuffered" is subsequently used in SI without further clarifications. For the experiments involving 4NC, this is also underlined in the caption of Fig. S5 illustrating 4NC+OH reaction: "Chromatograms illustrating the formation of phenolic products from 4-nitrocatechol+OH reaction in pure water."

**Lines 185-245**: Since the whole part is confused, I recommend shortening and writing the text more concisely explaining the mechanism with emphasize on the main formation pathways (shown in Fig. 2), and on your findings.

**Author's response:** We agree with this suggestion.

**Changes in the revised manuscript:** The discussion about the mechanism of reaction under investigation was revised and shortened from 1045 words (original) to 599 words (revised); please note that these numbers also include cited references.

**Line 282**: …"where it can undergo chemical and photochemical processing": What this statement refers to, clouds or wet aerosol, or both? From what has been written, one would conclude that the processes take place only in wet aerosols.

**Author's response:** We agree with this comment, this sentence was not well constructed.

**Changes in the revised manuscript:** These lines were revised as "The average measured Henry's law constant – $5\times104$ ($M\times atm-1$) – indicates that 4NP exists entirely in the aqueous phase in clouds but not in "wet" aerosols (Fig. S12) (Herrmann et al., 2015). Once dissolved in cloud water, 4NP can undergo chemical and photochemical processing; thus, the rates of the bleaching of the 4NP solution due to the reaction with OH were evaluated – Table 1. "

**Line 297**: Which two bleaching mechanisms: via OH reactions and via photolysis? From the results in Fig. 4, photolysis is not very effective.

**Author's response:** As presented in Fig.6 (previously Fig. 4), photolysis is only relevant (dominant) when the concentration of OH is sufficiently low, while the estimated $[OH]_{aq}$ strongly depending on the variety of clouds (x axis).

**Changes in the revised manuscript:** This line was revised as "As presented in Fig. 6, photolysis of 4NP may be relevant under realistic atmospheric conditions in urban and remote clouds, with the estimated mean concentration of $OH_{aq}$ lower than $1 \times 10^{-13}$ M (Herrmann et al., 2010)." Fig. 6 was revised slightly, x axis is now labeled as "Concentration of $OH_{aq}$ M".

**Referee 1; Technical corrections and comments to Supplemental material**

All references (in parentheses) have to be written from the earliest to the latest one according to the year of publication.

I suggest changing "absorptivity" with "absorption": in the title and throughout the manuscript: e.g., line 26: it should be "UV-Vis absorption"; line 54: "light absorption of aqueous particles", etc.

Line 54: The chemical and photochemical…..result (not results).

Line 76: Should be plural (…are strongly..).

Line 84: Should be plural (…were monitored..).

Line 90: Aqueous-phase reactor (here "aqueous-phase" is an adjective)

Line 105: Delete ", the instrument was"; it should be "and equipped with…"

Line 123: "was quantified" (or determined)

4-nitophenol (4NP ) can be written as 4NP, etc.

Base-e, base-10: it is no need to write all the time; it's obvious from the equations.

Fig.3: Data are presented…(plural)

Line 281: Instead of "resides" it's better "exists"

Line 297: …depending on [OH]

Page 15: Authors of the first reference are missing.

**Author's response:** All of the corrections listed were revised accordingly to the referee's suggestion. The list of references in the main text as well as in the SI was thoroughly reviewed, appropriate journal titles abbreviations, as well as missing authors' names, were added. Note also the use of the English language was improved by a professorial proofreader.

**Referee 2:**

**line 78:** Check for missing articles and the right use of singular and plural in the English

language. Here, it should read 'OH reactions'. Carefully revise the whole text.

**Line 123:** 'was quantified'

**Author's response:** The revised manuscript was proofread and corrected by a professional English proofreader (see the enclosed certificate).

**Referee 2:**

**Introduction:** Whilst is good to show that lots of references exist and are relevant to the present work, I feel this text is suffering from over-referencing which makes it very difficult to read. I would like to work with summaries such as "and references therein" so you do not need to cite each and every study on a one-by-one basis.

**Author's response:** We agree with this suggestion.

**Changes in the revised manuscript:** Where adequate, we have reduced the number of articles referenced in the introduction, some references that were cited only once (in the introduction) were removed. Note however that several references were added in the introduction following the suggestions and comments kindly provided by Referee 1.

**Referee 2:**

**Figure 3 (now Figure 4)**: Are these single experiments ?

**Author's response:** The uncertainty for these data points was derived from multiple UV-Vis and TOC measurements and was well below 3% for the individual data points. Consequently, uncertainty bars are smaller than data symbols in both Figs. 5 and S10 and are not visible. At the same time, we believe that making the data points smaller will result in decreased readability of these figures.

**Changes in the revised manuscript:** This clarification was added in section S8 (page 17 in the SI). x axis was renamed as "Amount of 4-nitrophenol reacted (%)".

**Referee 2:**

Fig.5 (previously Fig. 3) Why are the curves of (A) and (C) so scattered ?

**Author's response:** The positions of the curves shown in Fig. 5 reflect the influence of the pH at which the absorbance was measured on the absorption of the reaction mixture, which is higher under basic pH conditions. As discussed below Fig. 5 (formerly Fig. 3) such a result can be explained by higher ε molar absorption cross-sections of the nitrated phenols under investigation at basic pH (see also Fig. S4).

**Changes in the revised manuscript:** The discussion of the results presented in Fig. 5 (previously Fig. 3) was revised to clarify the presented conclusions.

**Referee 3:**

It is stated in the **abstract** "Hence, up to 65% of the organic carbon…". I can see in the conclusions where this statement comes from, but I don't see how there can be a hence leading to this coming from the sentence before it in the abstract. Please clarify where the 65% value is coming from.

**Author's response:** We agree with this comment, this conclusion was not well presented.

**Changes in the revised manuscript:** This sent ace was revised as: "Moreover, as inferred from the TOC measurements and the molar yields of the phenols formed, 65% of the organic carbon that remained in the aqueous solution was attributed to the non-aromatic products."

The **experimental** set-up uses two 254 lamps. It is stated that this allows for the formation of OH radicals from the added $H_2O_2$ and that the direct absorption of 4NP is avoided. Were any experiments done to confirm this? The supplemental refers to section 4.2, but there is no mention of photolysis only experiments with these lamps, or with the 254 nm lamps only, to show the possible role these might play in the rates.

**Author's response:** Extensive control experiments were carried out in order to confirm that the observed decrease in the concentration of 4NP was only due to reaction with the OH. In other words, only when UV irradiation and $H_2O_2$ were present together, the reaction was observed (see also section 2.7). Photolysis-only experiments (no $H_2O_2$ added) were indeed carried out at both pH=2 and 9 for all analytes – please refer to the first row in Table S3. As discussed in section S6, "As listed in Table S3, none of the phenols under investigation underwent direct photolysis or "dark" reactions with $H_2O_2$, within the time-scale of the experiments."

**In Figure 3**, you show the change in MAC vs. % 4NP reacted at different pH values. But, the data also shows different pH levels (colors). I believe that the pH is being adjusted after the aging process to generate this data. If this is the case, pleas clarify that in either the text or the caption. If not, please clarify what is being plotted.

**Author's response:** Referee is correct, the pH (2,6,8 or 9) refers to the pH at which absorption was measured, as described in section 2.4. The pH was adjusted in the UV-Vis cuvette, after the aging process, after taking the sample out of the photoreactor.

**Changes in the revised manuscript:** Line "The colors refer to the pH at which the absorbance was measured (section 2.4)" was added in the caption of Fig. 5 (previously Fig. 3).

**Page 13**, at the end of the section it is stated that "Also, because of a significant increase in the actinic flux at ….any "brown" products formed efficiently stabilize the $R_{abs}$ values through the course of the reaction…" I don't understand this statement. How do the brown product stabilize the $R_{abs}$ values? A little more clarity here would be helpful.

**Author's response:** We agree with this comment, this discussion was not well presented.

**Changes in the revised manuscript:** This discussion was revised as "The $R_{abs}$ values (Fig. 5C and 5D) decrease slower compared with the values of $MAC_{TOC}$ and become stable when the pH at which the absorbance was measured is less than 7 (see also Fig. S10). This is likely due to a red-shift of the $A_{max}$ of the reaction solution, likely connected to the red-shift of the 4NP and 4NCs absorbance following the increase in the pH (Fig. S4). As presented in Fig. S12, the actinic flux exhibits a significant increase when $\lambda > 400$ nm. Consequently, the BrC chromophores – products of reaction (I) – characterized by a strong absorbance above 400 nm, will contribute to the observed stabilization of the estimated $R_{abs}$ at pH<7."

**Cited references**

Aljawhary, D., Zhao, R., Lee, A. K. Y., Wang, C., and Abbatt, J. P. D.: Kinetics, Mechanism, and Secondary Organic Aerosol Yield of Aqueous Phase Photo-oxidation of α-Pinene Oxidation Products, J. Phys. Chem. A, 120, 1395-1407, 10.1021/acs.jpca.5b06237, 2016.

Bluvshtein, N., Lin, P., Flores, J. M., Segev, L., Mazar, Y., Tas, E., Snider, G., Weagle, C., Brown, S. S., Laskin, A., and Rudich, Y.: Broadband optical properties of biomass-burning aerosol and identification of brown carbon chromophores, Journal of Geophysical Research: Atmospheres, 122, 5441-5456, https://doi.org/10.1002/2016JD026230, 2017.

Chattopadhyay, A., Gierczak, T., Marshall, P., Papadimitriou, V. C., and Burkholder, J. B.: Kinetic fall-off behavior for the Cl + Furan-2,5-dione (C4H2O3, maleic anhydride) reaction, Physical Chemistry Chemical Physics, 23, 4901-4911, 10.1039/D0CP06402E, 2021.

Gierczak, T., Bernard, F., Papanastasiou, D. K., and Burkholder, J. B.: Atmospheric Chemistry of c-C5HF7 and c-C5F8: Temperature-Dependent OH Reaction Rate Coefficients, Degradation Products, Infrared Spectra, and Global Warming Potentials, The Journal of Physical Chemistry A, 125, 1050-1061, 10.1021/acs.jpca.0c10561, 2021.

Harrison, A. W., Waterson, A. M., and De Bruyn, W. J.: Spectroscopic and Photochemical Properties of Secondary Brown Carbon from Aqueous Reactions of Methylglyoxal, ACS Earth and Space Chemistry, 4, 762-773, 10.1021/acsearthspacechem.0c00061, 2020.

He, L., Schaefer, T., Otto, T., Kroflič, A., and Herrmann, H.: Kinetic and Theoretical Study of the Atmospheric Aqueous-Phase Reactions of OH Radicals with Methoxyphenolic Compounds, The Journal of Physical Chemistry A, 123, 7828-7838, 10.1021/acs.jpca.9b05696, 2019.

Herrmann, H., Hoffmann, D., Schaefer, T., Bräuer, P., and Tilgner, A.: Tropospheric Aqueous-Phase Free-Radical Chemistry: Radical Sources, Spectra, Reaction Kinetics and Prediction Tools, ChemPhysChem, 11, 3796-3822, https://doi.org/10.1002/cphc.201000533, 2010.

Kitanovski, Z., Shahpoury, P., Samara, C., Voliotis, A., and Lammel, G.: Composition and mass size distribution of nitrated and oxygenated aromatic compounds in ambient particulate matter from southern and central Europe – implications for the origin, Atmos. Chem. Phys., 20, 2471-2487, 10.5194/acp-20-2471-2020, 2020.

Richards-Henderson, N. K., Hansel, A. K., Valsaraj, K. T., and Anastasio, C.: Aqueous oxidation of green leaf volatiles by hydroxyl radical as a source of SOA: Kinetics and SOA yields, Atmospheric Environment, 95, 105-112, https://doi.org/10.1016/j.atmosenv.2014.06.026, 2014.

Witkowski, B., Al-sharafi, M., and Gierczak, T.: Kinetics and products of the aqueous-phase oxidation of β-caryophyllonic acid by hydroxyl radicals, Atmos. Environ., 213, 231-238, https://doi.org/10.1016/j.atmosenv.2019.06.016, 2019.

Witkowski, B., Chi, J., Jain, P., Błaziak, K., and Gierczak, T.: Aqueous OH kinetics of saturated C6–C10 dicarboxylic acids under acidic and basic conditions between 283 and 318 K; new structure-activity relationship parameters, Atmos. Environ., 267, 118761, https://doi.org/10.1016/j.atmosenv.2021.118761, 2021.

Zhao, S., Ma, H., Wang, M., Cao, C., and Yao, S.: Study on the role of hydroperoxyl radical in degradation of p-nitrophenol attacked by hydroxyl radical using photolytical technique, Journal of Photochemistry and Photobiology A: Chemistry, 259, 17-24, https://doi.org/10.1016/j.jphotochem.2013.02.012, 2013.

---

## Author Response (AR3)

We would like to thank the Editor and the anonymous referee for considering our manuscript as suitable for publication in ACP after minor revisions.

Editor's comments:

"The important question raised by referee 2 regarding the use of a linear fit to describe data that is clearly not following a linear trend in Fig. 2B has not been satisfactorily addressed. I see no discussion of the non-linear nature of the data in the manuscript, or why a linear fit was used for a portion of the data. This needs to discussed in the main text as well as mentioned in the figure caption. There should also be a discussion as to possible causes for the clear non-linear trend of observed nitrophenolate production. If you were to exclude the first data point for zero reaction you would get a different linear fit that would better describe that portion of the data. It may be reasonable to use a simple linear fit as an /approximation/ for some portion of the data but this needs to be clearly stated and discussed. You should also explain why you do not use a non-linear function that could likely well describe all of the data obtained. There is likely some interesting chemistry occurring that explains the non-linear nature of the data in Fig. 2B and a discussion of this would strengthen the manuscript. Please be sure to properly discuss this through further revisions to your manuscript."

Our responses to the comments provided together with the changes made in the revised manuscript are provided below.

**Referee 2 comment*:**

Figure 2B is using a linear fit to calculate the yield and a statistical analysis was carried out to show that this analysis was valid. However, by eye this is not a linear trend as there is clear curvature. Why were the points that are used selected to represent the linear portion? Please expand the analysis to show the variation found when a different range of initial points are used to provide uncertainty values for the assumptions that are made here

**Author's response:** To gain more insights into the underlying mechanisms of the 4NP+OH reaction a kinetic box-model was constructed.

The modeling results revealed that the non-linear trend observed in Fig. 2A was due to secondary reactions of the products formed (primarily 4NC) with the OH. This is an expected behavior of the plots derived via eq. I (Gierczak et al., 2021). For this reason, the yields (slopes) are derived from the initial portion of such plots via linear regression model; such a procedure is generally accepted for deriving realistic formation yields of a given product(s). Applying a different regression model and excluding the point of origin (0,0) from the regression analysis would certainly yield a better fit to the experimental data but results of such non-linear fitting cannot be connected with the mechanism of the reaction under investigation.

However, to reproduce the experimental data presented in Fig. 2B, the kinetic model had to be modified to include regeneration of 4NC from 4NC + OH with the yield of 0.5. The mechanistic implications of these findings are now discussed in section 3.1. A new, important conclusion obtained from the kinetic modeling is that the deprotonation of the precursor may enhance the disproportionation reaction of the nitrocyclohexadienyl-type radicals.

We agree that the behavior of the plots derived via eq. (I) was not sufficiently discussed in the manuscript.

**Changes in the revised manuscript:** Fig. 2 was revised, results of the linear fitting were removed. Instead, the modeled yields of 4NC generated by the kinetic box model are now included in Fig. 2A and Fig. 2B.

**Caption of Fig. 2** was revised as " The formation of phenolic products from the 4-nitrophenol (A, pH=2) and 4-nitrophenolate (B, pH=9) + OH reaction. The molar yields for 4NC were estimated from the initial sections of the plots via linear regression analysis (Fig. S4 and Table S3). Plots derived with eq. (I) for the products quantified are expected to curve during the course of the reaction because these molecules are also reactive towards OH. The lines for 4NC are results of kinetic modeling (Table S3 and Fig. S5)."

**Section S5** was added in the SI, containing results of the linear regression analysis, and the description and the results obtained from the kinetic box model. Modeling results are now referenced in the main text. The curving of the plots obtained via eq. (I) is reproduced by the box model developed, thereby confirming that their behavior is due to the secondary reactions of 4NC with the OH.

Results of the kinetic modeling was incorporated in the discussion connected with **Figs. 2 and 3** in the main text.

**Fig. 3** in the main text was revised due to minor formatting and editorial mistakes.

**Literature**

Gierczak, T., Bernard, F., Papanastasiou, D. K., and Burkholder, J. B.: Atmospheric Chemistry of c-C5HF7 and c-C5F8: Temperature-Dependent OH Reaction Rate Coefficients, Degradation Products, Infrared Spectra, and Global Warming Potentials, 125, 1050-1061, 10.1021/acs.jpca.0c10561, 2021.

Gierczak, T., Bernard, F., Papanastasiou, D. K., and Burkholder, J. B.: Atmospheric Chemistry of c-C5HF7 and c-C5F8: Temperature-Dependent OH Reaction Rate Coefficients, Degradation Products, Infrared Spectra, and Global Warming Potentials, 125, 1050-1061, 10.1021/acs.jpca.0c10561, 2021.

Herrmann, H., Schaefer, T., Tilgner, A., Styler, S. A., Weller, C., Teich, M., and Otto, T.: Tropospheric Aqueous-Phase Chemistry: Kinetics, Mechanisms, and Its Coupling to a Changing Gas Phase, Chem. Rev., 115, 4259-4334, 10.1021/cr500447k, 2015.

Gierczak, T., Bernard, F., Papanastasiou, D. K., and Burkholder, J. B.: Atmospheric Chemistry of c-C5HF7 and c-C5F8: Temperature-Dependent OH Reaction Rate Coefficients, Degradation Products, Infrared Spectra, and Global Warming Potentials, 125, 1050-1061, 10.1021/acs.jpca.0c10561, 2021.

Herrmann, H., Schaefer, T., Tilgner, A., Styler, S. A., Weller, C., Teich, M., and Otto, T.: Tropospheric Aqueous-Phase Chemistry: Kinetics, Mechanisms, and Its Coupling to a Changing Gas Phase, Chem. Rev., 115, 4259-4334, 10.1021/cr500447k, 2015.

Gierczak, T., Bernard, F., Papanastasiou, D. K., and Burkholder, J. B.: Atmospheric Chemistry of c-C5HF7 and c-C5F8: Temperature-Dependent OH Reaction Rate Coefficients, Degradation Products, Infrared Spectra, and Global Warming Potentials, 125, 1050-1061, 10.1021/acs.jpca.0c10561, 2021.